# A Robust Perturbation Strategy and Evaluation Benchmark for Accurate SHAP Attribution in Vision Models

## Abstract

Understanding the decisions of vision models is essential for transparency and trust. Shapley values provide a principled approach to feature attribution, yet their application to vision is hindered by perturbation strategies that either fail to exclude information or introduce artifacts. We propose the **Mean-Distance Perturbation (MDP)** strategy, a simple and deterministic method that replaces image regions with maximally dissimilar colors in RGB or grayscale space. Unlike blurring, inpainting, or uniform fills, MDP offers a safer perturbation mechanism that more reliably suppresses class-relevant evidence. We further present **SHAPEval**, the first vision benchmark with analytically derived, pixel-level Shapley ground truth. Although intentionally simple, SHAPEval provides a reproducible baseline for controlled attribution evaluation and establishes a foundation for future benchmarks on more complex image data. Experiments on SHAPEval and natural datasets show that MDP consistently outperforms standard perturbation strategies across multiple models. Together, MDP and SHAPEval deliver a reproducible starting point for safer and more reliable explainability in computer vision. By exposing how perturbation choices alter model confidence and attribution alignment, our work directly links explanation quality to the robustness of learned representations.

## 1 Introduction

Machine learning models are increasingly deployed in high-stakes domains, where trust depends not only on predictive accuracy but also on understanding why a model makes a given decision. Feature attribution methods aim to provide such understanding by quantifying the contribution of individual input components to a prediction. Among them, Shapley values Shapley & Corporation (1951) stand out for their strong theoretical foundation: they measure the marginal effect of each feature by comparing model outputs with and without that feature. In computer vision, realizing this principle requires carefully designed perturbations that remove visual evidence without introducing artifacts. If perturbations fail, by leaving residual signal or injecting out-of-distribution structure, the resulting attributions no longer reflect the model's true reliance on input features.

This problem is not merely cosmetic: perturbation choice directly influences the measured importance of regions, and thus how we interpret the robustness of the model's learned representations. Figure 1 illustrates standard perturbation strategies such as Gaussian blurring, inpainting, and uniform color replacement (UCR). These methods often fail to achieve faithful exclusion. For example, blurring preserves low-frequency information, while inpainting and UCR can add contextually misleading or globally biased patterns. Such issues motivate the need for a safer, more principled perturbation mechanism.

To this end, we propose the Mean-Distance Perturbation (MDP) strategy, which deterministically replaces regions with maximally dissimilar colors in RGB or grayscale space. MDP is simple, model-agnostic, and reduces residual signal while avoiding arbitrary priors. We further introduce SHAPEval, the first vision dataset with analytically derived, pixel-level Shapley ground truth. Instead of explaining a black-box model, SHAPEval uses a transparent linear function to generate

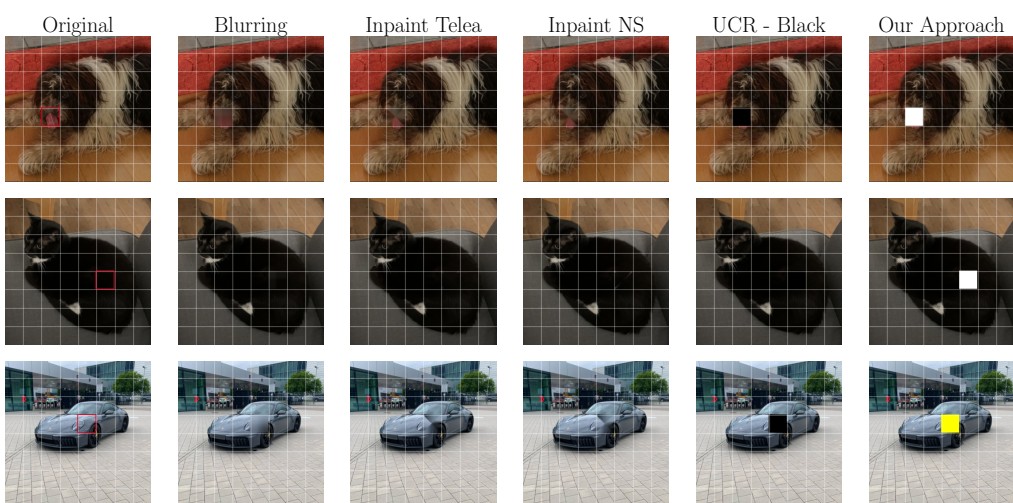

Figure 1: Examples of different perturbation strategies for SHAP value estimations

data with exact attributions, yielding a reproducible foundation for attribution research and enabling systematic study of how perturbation strategies interact with internal representations.

## 2 RELATED WORK

The SHAP (SHapley Additive exPlanation) Python library Lundberg & Lee (2017) implements various perturbation based methods for feature replacement: **Blurring**, which applies a Gaussian blur to the region $r$, reducing its detail but not fully eliminating its influence: **Inpainting**, which fills $r$ with content generated from surrounding areas but may introduce artifacts or contextually similar patterns, complicating interpretation and **Uniform Color Replacement**, which replaces $r$ with a solid color (e.g. black or gray), causing sharp discontinuities that may misalign with the model's learned data distribution. FastSHAP Jethani et al. (2022) enables single-pass SHAP estimation via a learned explainer network, offering efficiency. CXPlain Schwab & Karlen (2019) trains an auxiliary model to estimate feature importance by learning causal impacts, providing uncertainty quantification. These methods generate saliency maps for image data but rely on heuristic or learned approximations rather than ground truth contributions.

Building upon this SHAP values, Muschalik et al. introduced ShapIQ Muschalik et al. (2024), an open source Python package for efficiently computing Shapley Values (SVs) and Shapley Interactions (SIs) across diverse benchmark experiments span datasets such as Adult Becker & Kohavi (1996), Bike Sharing Fanaee-T (2013), and ImageNet Deng et al. (2009). For ImageNet, they utilized the mean RGB-room value (gray) for perturbation, illustrating a practical but limited perturbation approach for image data. Using the CIDDS-001 Ring et al. (2017) cybersecurity dataset (both tabular data), an inpainting approach based on denoising diffusion probabilistic models was proposed in Tritscher et al. (2024) to enhance Shapley value based explanations for anomaly detection tasks. The BONES benchmark Napolitano & Cagliero (2024) evaluates neural Shapley value estimation using datasets like ImageNet Deng et al. (2009) and Pet Parkhi et al. (2012a). It supports explainers such as DeepExplainer, GradientExplainer, FastSHAP, and ViT-Shapley Covert et al. (2023) (specific to transformer models). While comprehensive, BONES focuses on natural images, which lack well defined ground truth feature contributions. HarsanyiNet Chen et al. (2023) proposes an interpretable network capable of computing Shapley values in a single forward pass. Despite its efficiency, HarsanyiNet's scope is currently restricted to tabular datasets like Census Income Kohavi (1996) and Yeast Nakai (1991), necessitating further exploration for broader applicability.

Beyond Shapley values, gradient based approaches like Grad-CAM Selvaraju et al. (2019) compute the most important image regions influencing predictions in Convolutional Neural Networks (CNNs). Grad-CAM++ Chattopadhay et al. (2018) extends this by incorporating higher order

derivatives for finer granularity, while Smoothed-Grad-CAM++ Omeiza et al. (2019) reduces noise by averaging saliency maps across perturbed inputs. Eigen-CAM Muhammad & Yeasin (2020) leverages PCA on feature map activations for global feature relevance, which are not class specific by design. SHAP-CAM Zheng et al. (2022) combines gradient based methods with Shapley values, offering a hybrid explanation framework. Local Interpretable Model-Agnostic Explanations (LIME) Ribeiro et al. (2016) perturbs image regions to observe their impact on predictions, approximating the model locally using an interpretable surrogate (e.g. linear regression). While computationally lighter than SHAP, LIME is providing approximations rather than exact contributions.

According **SHAP ground truth datasets**, XAI-Bench Liu et al. (2021a) introduces synthetic datasets for computing conditional expected values, facilitating ground truth Shapley value evaluation and other explainability metrics. Its focus remains on tabular data, such as the Wine Quality Cortez & Reis (2009) and Forest Fire Cortez & Morais (2007) datasets, limiting its direct applicability to image data. In Jeyakumar et al. (2020), the authors conducted user studies via Amazon Mechanical Turk to evaluate preferences for explanation methods across domains (image, text, audio, sensory). However, this work is qualitative and lacks exact ground truth Shapley values for image pixels. Most studies leveraging datasets like Imagenet Deng et al. (2009) fail to establish true ground truth feature contributions for image data. For instance, determining the importance of specific features (e.g. a dog's nose, torso, or paws) remains inherently subjective. Artificial datasets like those in Miró-Nicolau et al. (2023) define ground truth Shapley values for simple geometric patterns, such as circles and squares, using binary values (0 or 1).

However, these datasets are limited to pattern level contributions and fail to operate in continuous value spaces. A critical gap exists in datasets offering explicitly defined, continuous ground truth feature contributions for image data. Developing datasets with explicit, granular feature contributions would significantly advance the evaluation of explainable AI methods, providing a robust foundation for assessing model interpretability. Unlike prior work, SHAPEval provides continuous, pixel level SHAP ground truth for vision tasks, a capability absent in BONES, ShapIQ, or XAI-Bench.

## 3 MDP: MEAN-DISTANCE PERTURBATION STRATEGY

The accuracy of Shapley based saliency attribution in vision models is fundamentally limited by the choice of perturbation function used to exclude image regions. Existing strategies such as Gaussian blurring, inpainting, or uniform color replacement (UCR) often fail to satisfy the key *removal* assumption of SHAP. That is, they either fail to remove semantically relevant content or introduce distributional artifacts, leading to biased attribution estimates. Let $x \in \mathbb{R}^{H \times W \times C}$ denote an input image segmented into $R$ disjoint regions $\{r_1, r_2, \ldots, r_R\}$. Following the SHAP framework, the perturbed image $\hat{x}(z')$ corresponding to a binary mask $z' \in \{0, 1\}^R$ is defined as:

$$\hat{x}(z') = \sum_{i=1}^{R} z_i' r_i + (1 - z_i')\bar{r}_i, \tag{1}$$

where $\bar{r}_i$ is a perturbation used to exclude region $r_i$. The corresponding Shapley value for region $r_i$ is computed as:

$$\phi_{r_i} = \sum_{\mathcal{R} \subseteq R \setminus \{i\}} \frac{|\mathcal{R}|!(|R| - |\mathcal{R}| - 1)!}{|R|!} \left[ f(h_x(\mathcal{R} \cup \{i\})) - f(h_x(\mathcal{R})) \right]. \tag{2}$$

In vision, standard perturbation methods violate two core desiderata: (1) complete semantic exclusion of region content, and (2) avoiding the introduction of strong artifacts or new structure. For example, blurring may leave low-frequency signals intact, while uniform black replacement $\bar{r}_i = 0$ can create strong global priors. These violations bias $\phi_{r_i}$. We therefore define a reliable perturbation strategy.

To achieve this, we propose the MDP strategy based on the mean color and its complement in the color space. The aim is to exclude the features encoded in $\mathcal{R}$ while ensuring the perturbation does not introduce unintended artifacts. Let $\mathcal{C}$ denote the color space of the image region, and let $p_i = [c_1, c_2, ..., c_d]$ represent the $d$-dimensional color vector of a pixel $p_i$. For a given region combination $\mathcal{R}$, we first compute the mean color $\mu_{\mathcal{R}}$, as a summary of its color information. The

mean color is calculated in the RGB-color space as:

$$\mu_{R,G,B} = \left( \frac{1}{|\mathcal{R}|} \sum_{p_k \in \mathcal{R}} R(x,y), \frac{1}{|\mathcal{R}|} \sum_{p_k \in \mathcal{R}} G(x,y), \frac{1}{|\mathcal{R}|} \sum_{p_k \in \mathcal{R}} B(x,y) \right),$$ (3)

where $|\mathcal{R}|$ denotes the number of pixels in the current region combination $\mathcal{R}$ and $p_k$ is the color value of the pixels in $\mathcal{R}$. This mean color $\mu_{\mathcal{R}}$ represents the overall color characteristics of the region. Next, we determine the most distant color $p*$ in the color space relative to $\mu_{\mathcal{R}}$. The distance metric $d$ is chosen based on the color space, ensuring it captures perceptual differences effectively. For the RGB color space, the Euclidean distance is commonly used:

$$d(p, \mu_{\mathcal{R}}) = \sqrt{\sum_{j=1}^{d} (p_j - \mu_{\mathcal{R},j})^2}$$ (4)

where $p_j$ and $\mu_{\mathcal{R},j}$ denote the $j$-th component of $p$ and $\mu_{\mathcal{R}}$, respectively. The final perturbation color $p*$ for the uniform color replacement is then identified as:

$$p* = \text{argmax}_{p \in \mathcal{C}} d(p, \mu_{\mathcal{R}})$$ (5)

which is the color furthest from the region's mean in the RGB space $\mu_{\mathcal{R}}$ in the chosen color space $\mathcal{C}$ and the current region combination $\mathcal{R}$. This color $p*$ is used to replace all pixel values in the region combination $\mathcal{R}$, generating a perturbed region $\hat{\mathcal{R}}$:

$$\hat{\mathcal{R}} = \{p*, p*, ..., p*\}.$$ (6)

Finally, the perturbed image $\hat{x}$ is reconstructed by combining the unaltered regions with the perturbed region:

$$\hat{x} = (x \setminus \mathcal{R}) \cup \hat{\mathcal{R}}$$ (7)

where $x \setminus \mathcal{R}$ denotes the original image excluding region $\mathcal{R}$ and $\hat{\mathcal{R}}$ is the perturbed region. This perturbation strategy ensures that the features within $\mathcal{R}$ are effectively excluded by replacing them with a uniform color that is maximally distinct from their mean color. This method dynamically adapts to the content of the region, while minimizing the risk of artifacts while remaining within the valid color range. By integrating this approach with SHAP's partitioning framework, we achieve robust and interpretable attributions for model predictions in computer vision tasks.

## 4 SHAPEVAL DATASET

To overcome the limitations of existing benchmarks described in Section 2, we propose a synthetic dataset designed to provide mathematically defined ground truth SHAP values to evaluate our proposed perturbation startegy.

### 4.1 APPROACH

The central idea behind the dataset is to reverse the standard application of SHAP by designing a known, interpretable model to generate the data. Rather than explaining a black box model, we use a transparent, known model to generate data with tractable SHAP values. This allows us to create this simple dataset in which we can calculate the SHAP values. This allows us to leverage the interpretability of the known model while ensuring that the derivation of SHAP values is tractable and precise. This approach ties back to the linear function definition in Lundberg & Lee (2017), ensuring consistency with the additive feature attribution framework. For this purpose, we utilize the grayscale color space, which forms a finite solution space defined as

$$\mathcal{L} = \{0, 1, 2, \dots, 255\}.$$ (8)

Here, $\mathcal{L}$ represents the set of all possible grayscale values, where each element is an integer and corresponds to a pixel intensity level ranging from black (0) to white (255). These integers are treated as real numbers for the purposes of the model's linear transformations, enabling the precise mathematical derivation of the dataset and its associated SHAP values.

## 4.2 CLASS ASSIGNMENT

The binary class label for each pixel value is determined by applying a threshold to the model output $M(x)$. The threshold $t$ is chosen to split the grayscale space into two regions corresponding to the two classes. For this study, we use a threshold value of $t = 127$, leading to the classification rule

$$\hat{y} = \begin{cases} 0, & \text{if } M(x) < t, \\ 1, & \text{if } M(x) \geq t. \end{cases} \tag{9}$$

This deterministic classification ensures a precise and reproducible mapping of pixel values to class labels.

## 4.3 GENERATION OF GROUND TRUTH SHAP VALUES

To evaluate feature contributions in this dataset, we compute SHAP values for each pixel intensity $I(x, y)$. These values quantify the contribution of each feature to the model's prediction. In the context of this dataset, the pixel intensity distance serves as the feature: how far a given intensity value deviates from the extremes of 0 or 255 determines its contribution to the prediction. The threshold or so called zero point $Z$, representing the midpoint of the grayscale range, is located at 127. At this point, the pixel is equally distant from both extremes, making its net contribution neutral with respect to both classes. With a SHAP value of zero. The zero point $Z$ is mathematically defined as:

$$z = \frac{max(\mathcal{L}) - min(\mathcal{L})}{2} \tag{10}$$

Using this zero point, the SHAP values for Class 0 and Class 1 are computed as:

$$\phi_{GT,0}(x) = z - x \qquad \phi_{GT,1}(x) = x - z. \tag{11}$$

These formulas ensure that the SHAP values represent the deviation of each pixel intensity from the neutral contribution point $z$, with separate contributions calculated for Class 0 and Class 1. To ensure consistency and compatibility with downstream processing of any arbitrarily model $f$, the computed SHAP values are normalized to the range $[-1, 1]$ by dividing by 255, the maximum pixel intensity. The normalized SHAP values for Class 0 and Class 1 are thus given by:

$$\phi_{GT,0}(x) = \frac{z - x}{255} \qquad \phi_{GT,1}(x) = \frac{x - z}{255} \tag{12}$$

This normalization step aligns the SHAP values with a consistent range, allowing for effective comparison and interpretation. In addition to normalizing the SHAP values, the output of the model $M(x)$ is also normalized to the same range $[-1, 1]$. This ensures compatibility and comparability between the ground truth SHAP values $\phi_{GT}$ and the predicted SHAP values $\hat{\phi}$, facilitating a robust evaluation framework.

## 4.4 DATASET CONSTRUCTION

To simulate practical conditions, we construct grayscale images of size $256 \times 20$ by sampling pixel intensities from either the full solution space $\mathcal{L}$ or a restricted subspace $\mathcal{T} \subset \mathcal{L}$. The constrained range $\mathcal{T} \in [50, 200]$ serves as a practical approximation, ensuring realistic variation while maintaining interpretability and reflecting situations with limited data availability. Each image is processed by a linear model $M(x)$, which computes the sum of pixel values, and binary class labels are assigned using a fixed threshold $t = 127$ as defined in Equation 9. For both $\mathcal{L}$ and $\mathcal{T}$, the data is split into training (70%), validation (10%), and test (20%) sets. Due to the linearity of $M$, exact pixel-level SHAP values can be derived analytically, yielding a fully labeled dataset where every instance is paired with its ground truth attribution map.

## 4.5 WHY SHAPEVAL MATTERS

SHAPEval introduces a new perspective on attribution evaluation: instead of explaining a fixed black-box model on natural images, we reverse the problem by generating data from an interpretable linear model. Because the value function is transparent, every pixel receives an exact Shapley value during generation. This makes SHAPEval the first vision dataset with continuous, pixel-level SHAP

ground truth, enabling attribution methods to be evaluated against analytical truth rather than heuristics or subjective labels. Unlike prior benchmarks such as BONES (region-level), XAI-Bench (tabular), or AIXI (binary object-level), SHAPEval provides tractable pixel-level attributions in vision. The simplicity of the construction is deliberate: it yields full tractability while providing a foundation that can be extended to richer data domains. With this design, different perturbation methods can be compared truthfully and reproducibly under mathematically defined conditions.

## 5 EXPERIMENTAL RESULTS

In this section, we evaluate the proposed perturbation strategy and compare explainers using SHAPEval and three real-world datasets. For perturbation evaluation, we trained a lightweight CNN with two $3\times3$ convolutional blocks on the synthetic dataset $M(\mathcal{T})$ (Section 4). Models were optimized with cross-entropy loss and Adam ($lr = 10^{-4}$), trained for up to ten epochs with batch size 32 and early stopping. The network processes $256 \times 20$ grayscale images and achieved 95% validation accuracy on $M(\mathcal{T})$ and 93% on $M(\mathcal{L})$. On test sets, $M(\mathcal{T})$ reached 96.5% accuracy (loss 0.12) and $M(\mathcal{L})$ 94.8% (loss 0.15), confirming the models reliably learned the pixel to class mapping.

### 5.1 COMPARISON OF PERTURBATION METHODS

To compare the different implemented perturbation methods (see Section 2), we generated an artificial image containing all possible grayscale values $I \in \{0, 255\}$ (Figure 1) and computed normalized SHAP values for each class. As the pixel intensity moves closer to either extreme (0 or 255), the SHAP attribution increases in favor of the corresponding class (0 or 1), approaching a value of 1. Conversely, intensities near the midpoint contribute negatively, indicating support for the opposite class. This image served as the baseline for our perturbation comparison. We applied all methods to this image tested on the trained model. To evaluate the methods, we calculated the absolute residual

$$\epsilon = \sum \phi_{GT} - \hat{\phi} \tag{13}$$

with $\hat{\phi}$ as the predicted SHAP value for each class, along with the mean $\bar{\epsilon}$ and standard deviation (Table 1).

| Method | Model M($\mathcal{T}$) | | | | Model M($\mathcal{L}$) | | | |
| --- | --- | --- | --- | --- | --- | --- | --- | --- |
| | $\sum \epsilon_0$ | $\sum \epsilon_1$ | $\bar{\epsilon}_0$ | $\bar{\epsilon}_1$ | $\sum \epsilon_0$ | $\sum \epsilon_1$ | $\bar{\epsilon}_0$ | $\bar{\epsilon}_1$ |
| Blurring $k = 5$ | 2714.87 | 2958.44 | 0.53 | 0.58 | 2965.05 | 2814.80 | 0.58 | 0.55 |
| Blurring $k = 15$ | 3061.53 | 2972.86 | 0.60 | 0.58 | 2580.96 | 2547.54 | 0.50 | 0.50 |
| Blurring $k = 51$ | 2216.37 | 2173.35 | 0.43 | 0.42 | 1588.99 | 1599.10 | 0.31 | 0.31 |
| Inpaint Telea | 3109.34 | 3093.23 | 0.61 | 0.60 | 2849.65 | 2805.00 | 0.56 | 0.55 |
| Inpaint NS | 2835.23 | 2860.24 | 0.55 | 0.56 | 2657.54 | 2651.32 | 0.52 | 0.52 |
| UCR-Black | 1528.19 | 1634.69 | 0.30 | 0.32 | 1596.15 | 1738.48 | 0.31 | 0.34 |
| UCR-White | 1556.49 | 1677.56 | 0.30 | 0.33 | **1171.59** | 1307.46 | **0.23** | 0.26 |
| MDP | **1098.23** | **1089.00** | **0.21** | **0.21** | **1052.85** | **1045.94** | **0.21** | **0.20** |
| MedianDP | 2025.49 | 1984.58 | 0.39 | 0.38 | 1753.49 | 1720.32 | 0.34 | 0.33 |

Table 1: Residual comparison of the different perturbation methods using the two trained models on our proposed SHAPEval data. UCR=Uniform Color Replacement

The results show that perturbation strategies differ substantially in attribution quality. Blur and inpainting based methods yield comparatively high residuals, with error increasing for larger kernels or more aggressive filling, reflecting that such replacements often leave residual class evidence or introduce strong artifacts. In contrast, uniform color replacement with extreme values (black/white) performs better, and our proposed MDP achieves the lowest residuals across both models, yielding the closest alignment with ground truth Shapley values.

For completeness, we also report a median-based variant (MedianDP), which replaces regions with the most distant median color instead of the mean. While more robust to outliers, it performs slightly

worse than MDP in practice. This trend aligns with Hooker et al. Hooker et al. (2019), who argued that attribution evaluation suffers when perturbations introduce out-of-distribution artifacts: the more OOD the replacement, the less reliable the explanation. SHAPEval quantitatively confirms this claim at pixel level, highlighting the importance of artifact averse, structure free perturbations.

## 5.2 EXPLAINER COMPARISON BY SHAPEVAL

We evaluated our proposed perturbation strategy and compared three SHAP explainers and three gradient based methods. GradientExplainer and DeepExplainer were tested with ten background images. In addition to the standard MDP variant, we also tested a version of the perturbation strategy that uses median based action selection to reduce sensitivity to outliers. All explainers were applied to our trained model using the test image from Section **??**, with $topk = 4$, batch size 50, and $n_{evals} = 10000$. Figure 2 shows the prediction results. PartitionerExplainer and GradientExplainer produced similar outputs due to analogous implementations, but struggled near the SHAP zero point and showed bias toward Class 0. Table 2 summarizes the quantitative comparison.

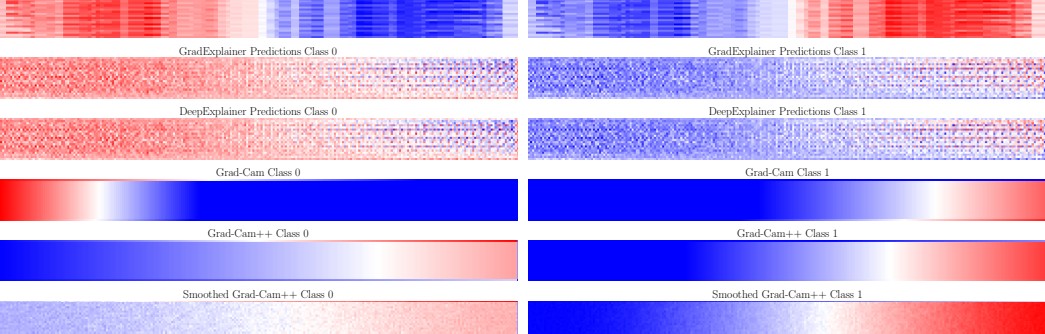

Figure 2: Ground truth SHAP values and the predictions of the different explainer

PartitionerExplainer achieved the lowest residual errors ($\epsilon_0$ and $\epsilon_1$), but exhibited high variability due to its reliance on Markov Decision Processes. GradientExplainer and DeepExplainer yielded identical statistics, reflecting their shared background generation mechanism. We also tested Grad-CAM and its extensions, using 50 noise examples (noise std: 0.1). As these methods only detect positive contributions, error analysis was limited to positive SHAP values. Grad-CAM showed the lowest error, while its extensions exhibited higher error, particularly on Class 0, due to complex gradient weighting. Eigen-CAM was excluded due to its PCA based attribution matrix, which prevents direct comparison with continuous SHAP values. FastSHAP and CXPlain, which rely on surrogate models, failed to yield robust results on the small training set (256 samples). LIME was also excluded, as its binary output lacks the resolution needed for fine grained attribution comparison. In summary, while all explainers provided insights into feature importance, their performance varied considerably depending on implementation details, baseline handling, and class sensitivity.

| Explainer | $\epsilon_0$ | $\epsilon_1$ | $\bar{\epsilon}_0$ | $\bar{\epsilon}_1$ | $\sigma(\epsilon_0)$ | $\sigma(\epsilon_1)$ |
|---|---|---|---|---|---|---|
| Partitioner* | 1233 | 1198 | 0.24 | 0.23 | 0.19 | 0.19 |
| Gradient | 2231 | 2295 | 0.44 | 0.45 | 0.28 | 0.29 |
| Deep | 2231 | 2295 | 0.44 | 0.45 | 0.28 | 0.29 |
| Grad-CAM | 1596 | 1545 | 0.31 | 0.30 | 0.29 | 0.30 |
| Grad-CAM++ | 3672 | 1708 | 0.71 | 0.33 | 0.46 | 0.28 |
| Smoothed-GC | 3184 | 1983 | 0.72 | 0.38 | 0.42 | 0.28 |

Table 2: Residual results of the different SHAP explainer on our SHAPEval data. *Utilized with MDP

## 5.3 REAL WORLD EXPERIMENTS

In natural image datasets, ground truth Shapley values are unavailable, making direct evaluation of attribution quality infeasible. Following Hooker et al. Hooker et al. (2019), model degradation offers a principled proxy: if a perturbation method removes class relevant evidence while minimizing out-of-distribution artifacts, the classifier's confidence should decrease in proportion to the true importance of the excluded region. Comparing confidence drops across perturbation strategies therefore reveals which method better isolates predictive content. While degradation cannot fully disentangle feature removal from distribution shift, its consistent adoption in attribution benchmarks Hooker et al. (2019); Covert et al. (2022) establishes it as a robust comparative criterion for real world evaluation. Combined with SHAPEval, which provides analytical ground truth in controlled settings, this two tier evaluation enables rigorous assessment of perturbation alignment across both synthetic and natural domains. To demonstrate real world effectiveness, we apply this protocol to pretrained image classification models on three standard benchmarks: ImageNet-S50 Gao et al. (2022), Oxford-IIIT Pets Parkhi et al. (2012b), and Oxford Flowers-102 Nilsback & Zisserman (2008). This analysis complements our synthetic results and directly addresses the question of whether MDP generalizes to high resolution, natural image distributions. We quantify model sensitivity to region removal via the prediction drop

$$\Delta f = f(x) - f(\hat{x}), \tag{14}$$

where $f(x)$ is the model's confidence on the original input and $f(\hat{x})$ is the confidence after perturbation. Larger values of $\Delta f$ indicate more effective feature exclusion, signifying that the removed region carried greater influence on the model's decision. We apply the perturbation methods to the object region only. Each perturbed image $\hat{x}$ is generated by applying one of the evaluated strategies. To maintain comparability with the SHAP baseline, we apply all perturbation methods on a fixed grid with 154 cells, which corresponds to approximately 1000 evaluations in the SHAP framework in. Models are not fine tuned; we use standard pretrained networks from torchvision maintainers & contributors (2016). Table 3 reports the average prediction collapse $\overline{\Delta f}$ across all samples per model and dataset.

| Model | MDP | MedianDP | UCR Black | UCR White | Blurring | Inpaint Telea | Inpaint NS |
|---|---|---|---|---|---|---|---|
| Imagenet-S50 | | | | | | | |
| ResNet18 | **0.0072** | **0.0072** | 0.0061 | 0.0059 | 0.0021 | 0.0022 | 0.0022 |
| ResNet34 | **0.0085** | **0.0085** | 0.0072 | 0.0074 | 0.0018 | 0.0021 | 0.0021 |
| ResNet50 | **0.0065** | **0.0065** | 0.0058 | 0.0060 | 0.0013 | 0.0017 | 0.0017 |
| MobileNetV2 | **0.0074** | **0.0074** | 0.0062 | 0.0066 | 0.0022 | 0.0027 | 0.0027 |
| VGG16 | **0.0080** | **0.0080** | 0.0072 | 0.0069 | 0.0021 | 0.0031 | 0.0033 |
| AlexNet | **0.0114** | **0.0114** | 0.0078 | 0.0094 | 0.0010 | 0.0016 | 0.0017 |
| Oxford-IIIT Pets | | | | | | | |
| ViT-B16 | **0.0016** | **0.0016** | 0.0001 | 0.0003 | 0.0010 | 0.0012 | 0.0011 |
| Oxford Flowers-102 | | | | | | | |
| ResNet50 | **0.0053** | 0.0052 | 0.0047 | 0.0047 | 0.0008 | 0.0002 | 0.0002 |
| ViT-B16 | **0.0042** | **0.0042** | 0.0026 | 0.0026 | 0.0003 | 0.0004 | 0.0004 |
| Swin | **0.0010** | **0.0010** | 0.0007 | 0.0007 | 0.0005 | 0.0002 | 0.0002 |

Table 3: Average prediction confidence drop after perturbing object regions across different datasets and model architectures: ResNet He et al. (2015), MobileNetV2 Sandler et al. (2019), VGG16 Simonyan & Zisserman (2015), AlexNet Krizhevsky et al. (2012), ViT Dosovitskiy et al. (2021), Swin Liu et al. (2021b)

In all tested settings, MDP achieves the largest confidence drop, consistently outperforming classical perturbation methods. On ImageNet-S50, for example, MDP and MedianDP yield the highest $\overline{\Delta f}$ across all models with up to 5–10× larger collapse compared to blurring or inpainting. Interestingly, the margin between MDP and UCR (uniform color replacement) grows in deeper or transformer based models (ViT, Swin). On Oxford Flowers-102 and Pets, MDP consistently induces the

largest attribution aware collapse across both models, validating its generality across architectures and datasets. These results highlight two key strengths of our method.

First, MDP better satisfies the "removal" assumption underlying SHAP by reliably eliminating informative content in the perturbed region without adding new structure. Second, unlike fixed color or generative inpainting methods that may introduce distributional artifacts or insufficient suppression, MDP dynamically adapts to region statistics while remaining deterministic and model agnostic. In summary, this large scale benchmark confirms that MDP is the most effective feature exclusion strategy for attribution evaluation, inducing the strongest and most consistent prediction drop across datasets and model types.

## 6  DISCUSSION AND LIMITATIONS

Prediction drop ($\Delta f$) is a widely used proxy for attribution evaluation, but it conflates two factors: removal of class relevant evidence and unintended out-of-distribution (OOD) artifacts. Extreme replacements may lower confidence simply because the perturbed image becomes implausible, so $\Delta f$ cannot be read as a pure measure of attribution quality. SHAPEval addresses this by providing analytical ground truth Shapley values under a controlled linear model, enabling attribution evaluation independent of distribution shift. While SHAPEval uses small grayscale images, this simplicity is by design: it enables exact attribution ground truth, which is impossible to obtain on complex data. The goal is not to replicate ImageNet-scale diversity, but to provide a reproducible baseline that complements large scale evaluations.

On natural images, we retain $\Delta f$ as a complementary stress test in line with prior work Hooker et al. (2019); Covert et al. (2022), but interpret it cautiously. Within this paradigm, MDP emerges as the most reliable among existing strategies: it removes evidence more effectively than blurring or inpainting while avoiding the arbitrary priors of fixed color fills. OOD effects remain a fundamental limitation of perturbation based explainability, but MDP represents a practical advance. Future work should explore distribution aware perturbations, retrain-based protocols (ROAR/KAR), or generative inpainting to further separate true feature removal from artifact-driven effects. Generative perturbations (e.g. diffusion based inpainting like Rout et al. (2023); Afshar et al. (2024); Kim et al. (2024)) are promising but orthogonal. They introduce new structure, whereas MDP is deliberately structure free, making it a safer baseline for attribution evaluation.

## 7  CONCLUSION

This paper tackles a key gap in SHAP based explainability for vision through two complementary contributions. First, we propose Mean-Distance Perturbation (MDP), a simple and deterministic strategy that replaces image regions with maximally dissimilar colors in RGB or grayscale space. Unlike blurring, inpainting, or uniform fills, MDP provides a safer perturbation mechanism that more reliably removes informative content without introducing strong artifacts. Second, we introduce SHAPEval, the first vision benchmark with analytically derived, pixel level Shapley ground truth. Designed to be simple but fundamental, SHAPEval enables rigorous and reproducible evaluation of attribution methods and forms a baseline for future benchmarks on more complex images. Our experiments show that MDP consistently outperforms standard perturbations on SHAPEval and natural datasets—including ImageNet-S50, Oxford Flowers-102, and Oxford-IIIT Pets—achieving stronger and more consistent prediction collapse across models. At the same time, the T vs. L experiments reveal an inherent limitation: when the evaluation distribution shifts, attribution quality degrades, echoing concerns raised in prior work. While MDP makes perturbation safer and more effective, perfect reliability requires knowledge of the training distribution, pointing to future directions such as distribution aware perturbations and generative replacements. In summary, we contribute a new perturbation method and the first tractable dataset for pixel level ground truth in vision. Together, MDP and SHAPEval establish a reproducible foundation and open the path toward more complex, distribution aware explainability benchmarks.

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

## A APPENDIX

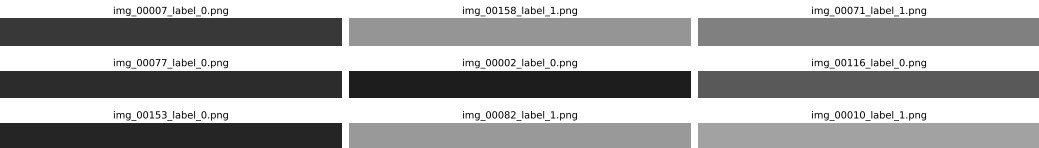

Figure 3: Randomly selected instances from the SHAPEval dataset. Each $256 \times 20$ grayscale image is paired with analytically derived pixel-level Shapley values. The dataset supports controlled benchmarking of attribution methods by ensuring both precise class labels and exact feature contributions, bridging the gap between theoretical SHAP definitions and vision-specific evaluation.

## LLM USAGE DISCLOSURE

We used a large language model (GPT-5 Thinking via ChatGPT) strictly for grammar and style polishing of text written by the authors. The model did not generate new scientific content, claims, analysis, figures, tables, or references. All technical content and conclusions are the authors' own, and the authors verified every change. Any errors remain our responsibility.

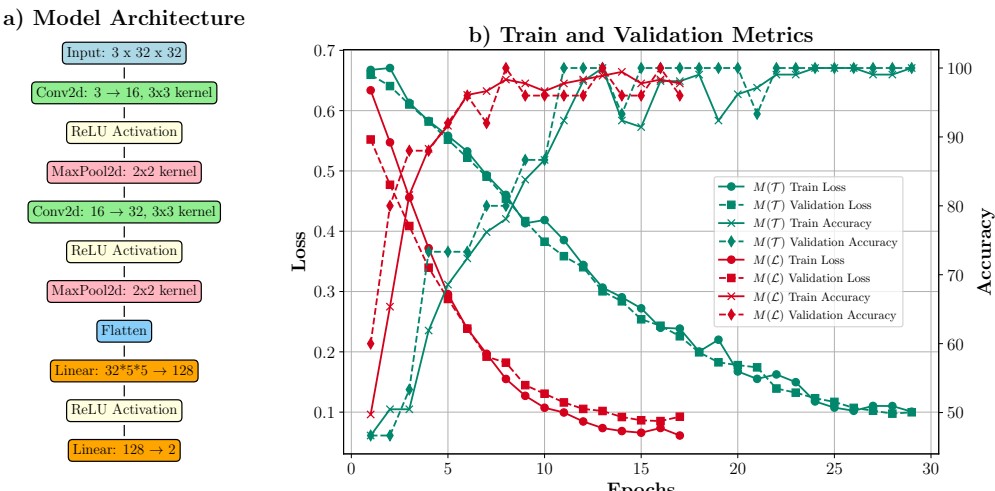

Figure 4: Architecture and training behavior of model $M$. The network consists of two $3\times3$ convolutional blocks trained with cross-entropy loss on SHAPEval images ($256 \times 20$ pixels). Validation accuracy reached 95% and test accuracy 96.5%, confirming that $M$ reliably learns the pixel-intensity-to-class mapping required for ground-truth attribution evaluation.

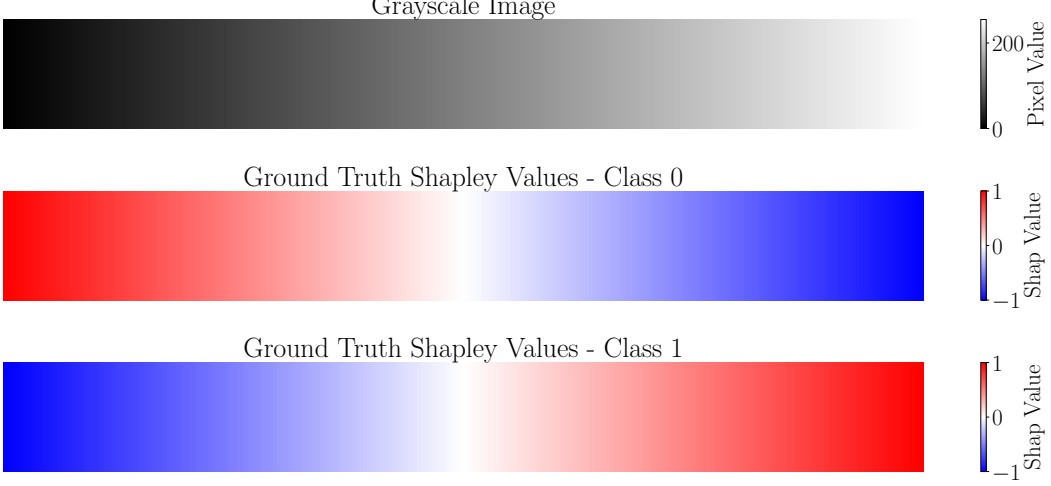

Figure 5: Ground-truth SHAP attribution in SHAPEval. The synthetic setup defines exact pixel-level contributions under a linear additive model, enabling analytical computation of Shapley values. This provides a unique benchmark for evaluating perturbation strategies with mathematically defined ground truth rather than qualitative saliency visualizations.

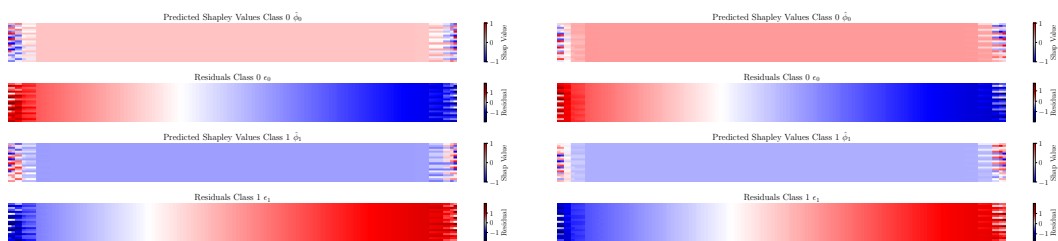

Figure 6: Evaluation of Gaussian blurring with kernel size $k = 5$: $T$ (left) vs. $L$ (right).

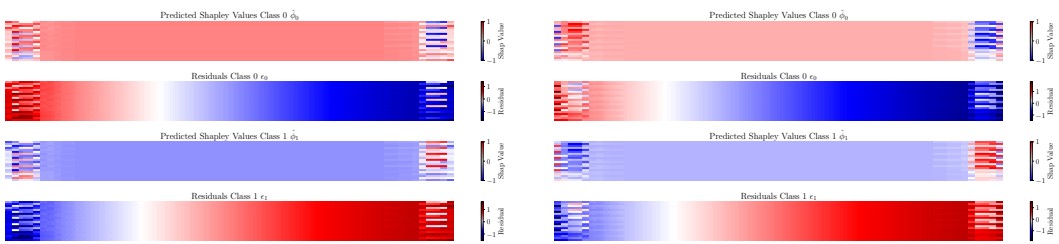

Figure 7: Gaussian blurring with kernel size $k = 15$: $T$ (left) vs. $L$ (right).

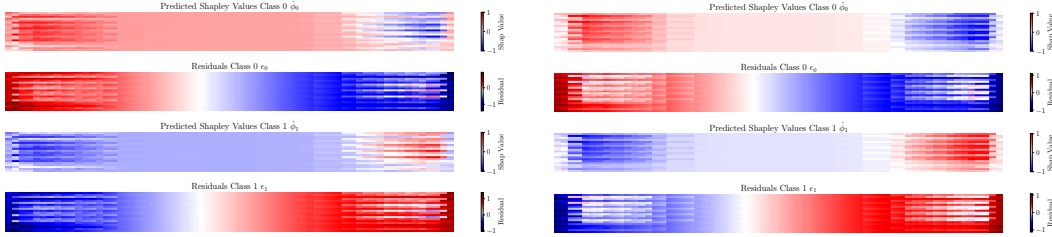

Figure 8: Gaussian blurring with kernel size $k = 51$: $T$ (left) vs. $L$ (right).

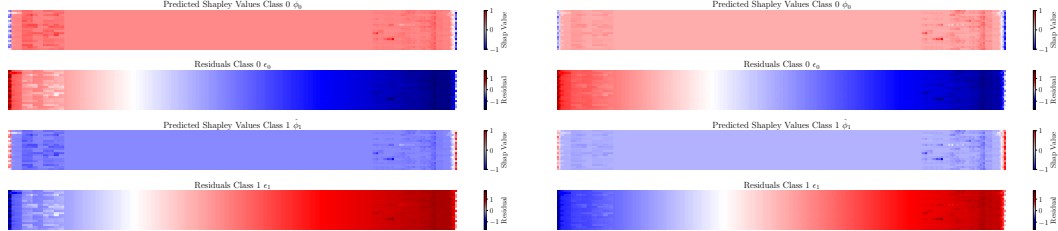

Figure 9: Inpainting with Telea: $T$ (left) vs. $L$ (right).

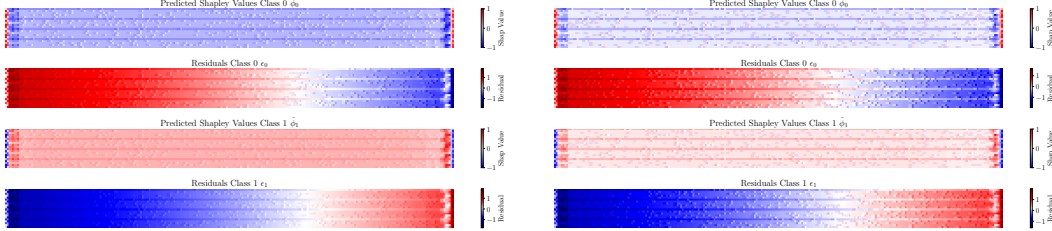

Figure 10: Inpainting with Navier–Stokes: $T$ (left) vs. $L$ (right).

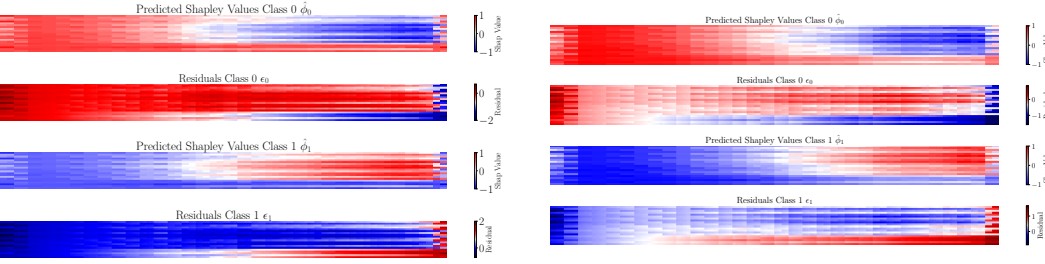

Figure 11: Uniform data perturbation (black): $T$ (left) vs. $L$ (right).

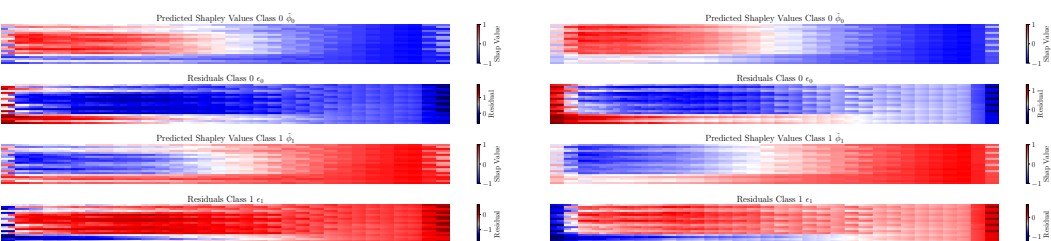

Figure 12: Uniform data perturbation (white): $T$ (left) vs. $L$ (right).

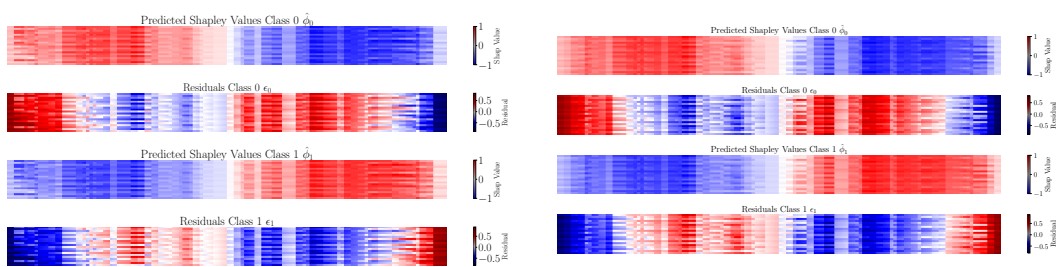

Figure 13: Mean-Distance Perturbation (MDP): $T$ (left) vs. $L$ (right).

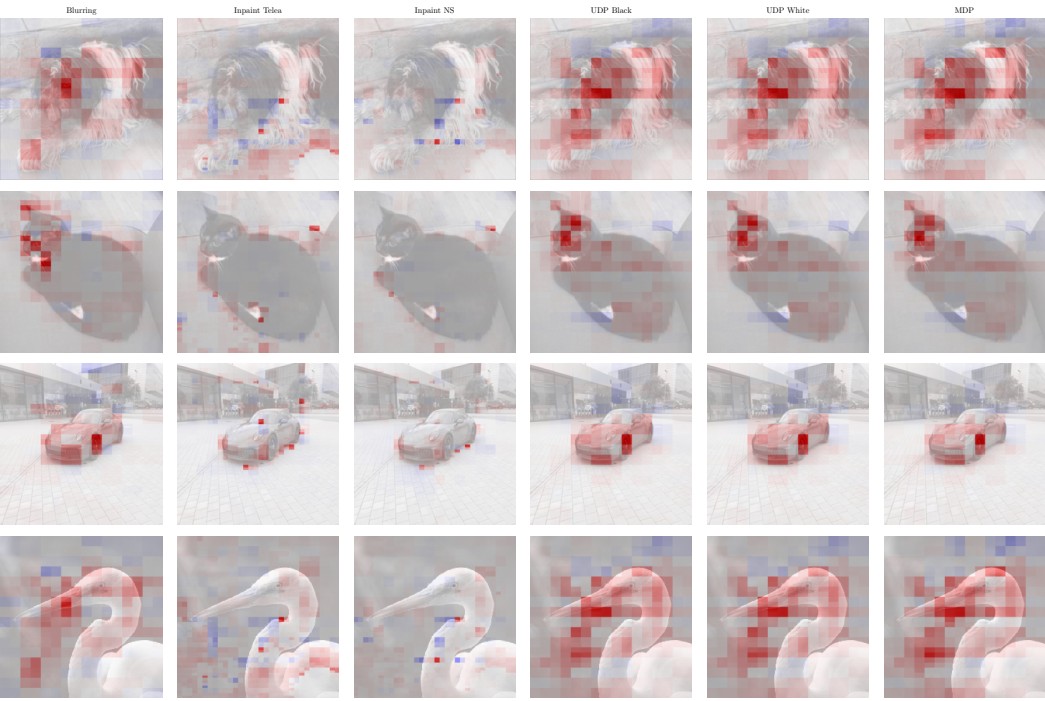

Figure 14: MobilNetV2 attribution comparison on own examples and original Lundberg & Lee (2017) example using different perturbation strategies (blurring, inpainting, uniform color replacement, and MDP). Shown in the style of the original Lundberg & Lee (2017) documentation, this example illustrates how MDP produces stronger and more consistent evidence removal than standard baselines.

