# OpenReview forum: "A Robust Perturbation Strategy and Evaluation Benchmark for Accurate SHAP Attribution in Vision Models"
_ICLR.cc/2026/Conference — Submitted to ICLR 2026_

### Official Review · Reviewer_Ahz7 · 2025-10-27

**Soundness:** 2
**Presentation:** 3
**Contribution:** 1
**Rating:** 2
**Confidence:** 4

**Summary:**

The paper addresses the problem of selecting an appropriate baseline for SHAP-based explanations and evaluating attribution methods. It introduces the Mean-Distance Perturbation (MDP) strategy, which defines the baseline image using colors that are maximally dissimilar in RGB or grayscale space. For evaluation, the paper proposes a linear model and a synthetic dataset where the classification process is fully transparent, allowing the computation of ground-truth Shapley values for comparison with estimated ones. The paper evaluates the proposed baseline strategy both on this controlled linear setup and on real image data, demonstrating the effectiveness of the MDP approach for improving the reliability of SHAP-based explanations.

**Strengths:**

- The paper is clearly written and easy to follow.
- It addresses an important problem in evaluating explanations and provides a well-motivated discussion of the limitations of existing baseline image strategies.
- The proposed method is tested across different datasets and models.

**Weaknesses:**

- **Insufficient evaluation for an incremental idea.** While the idea of MDP is interesting and well motivated, it represents a relatively incremental contribution. In my view, such a contribution would require a very extensive evaluation to convincingly demonstrate its advantages over existing baseline strategies, which is not sufficiently provided in the paper.
- **The novelty of the evaluation seems somewhat overstated.** While the paper presents an interesting approach, similar works have already explored the estimation of ground-truth importances in images using principled methods. Claims such as "the first vision benchmark with analytically derived, pixel-level Shapley ground truth" (L. 020), therefore, appear to be exaggerated. For instance, *"Scrutinizing XAI using linear ground-truth data with suppressor variables"* employs a simple data-generation process and a linear model to derive ground-truth importances, and *"FunnyBirds: A Synthetic Vision Dataset for a Part-Based Analysis of Explainable AI Methods"* uses image-space interventions on a synthetic dataset to obtain ground-truth importances on pixel groups. The paper should discuss these and other related approaches and clearly articulate the distinct contribution and necessity of the proposed method.
- **The evaluation relies on assumptions that may not hold.** I am not fully convinced by the evaluation design. It is unclear why a discrete model that applies thresholding would yield different ground-truth Shapley values depending on the distance to $z$. Wouldn’t an input with $M(x) = 128$ be classified identically to one with $M(x) = 255$ if both fall on the same side of the threshold? Moreover, the choice of the threshold $t$ itself may strongly influence the evaluation results. With $t$ placed at the midpoint of the pixel range, the use of maximally dissimilar pixels as a baseline, as proposed in MDP, naturally appears advantageous. However, if $t$ were near the boundaries (e.g., 1 or 254), a zero baseline might actually yield better performance.
- **Evaluation on real images.** There exist several well-established evaluation protocols for feature attribution that are more commonly used than the one proposed in Section 5.1 (for example, the incremental deletion protocol). The paper would be stronger if it incorporated such established evaluation methods. Furthermore, it would be valuable to examine how other attribution approaches that rely on a baseline image, such as Integrated Gradients, perform when combined with the proposed MDP strategy. More generally, it would be useful to compare the proposed method against other established attribution techniques on real-world datasets, similar to the analysis presented in Table 2.

**Minor weaknesses**

- In L. 335, there is wrong reference (??).

**Questions:**

- In Table 3, does the inpainting method refer to the approach used to generate the attributions, the one used to measure the prediction confidence drop, or both?
- It would be valuable to evaluate how MDP performs in combination with different attribution methods across a broader range of established evaluation protocols, and how these results compare to state-of-the-art attribution methods.
- What is the need for the proposed evaluation strategy, given that other principled and widely accepted evaluation approaches for attribution methods already exist?

I thank the authors for their effort and look forward to reading the rebuttal.

---

> ### Author Response · Authors · 2025-11-27
>
> We appreciate the reviewer’s detailed comments. The comments precisely highlight the aspects that needed clarification and the points raised proved extremely helpful for improving the clarity and evaluation depth of the paper. The following response directly addresses each concern.
>
> **Weakness 1: Incremental contribution and evaluation scope**
> The reviewer is absolutely right that a focused contribution such as MDP requires solid empirical support. The goal is to provide a perturbation baseline that is simple, deterministic, and broadly usable across explainers. The evaluation already spans analytical ground truth on SHAPEval, diverse perturbation strategies, two data regimes, and three real-world datasets with nine architectures. To meet the reviewer's expectations even more closely, the final version will include incremental deletion/insertion curves and integrate MDP with Integrated Gradients and additional SHAP explainers. This directly addresses the request for a more extensive evaluation and strengthens the empirical case for MDP as a general-purpose baseline.
>
> **Weakness 2: Novelty relative to Scrutinizing XAI and FunnyBirds**
> Thank you for this pointer! We agree that our positioning needs to be more precise. We will explicitly cite and discuss both works and soften the “first” claim. Scrutinizing XAI derives ground-truth importances for linear models on low-dimensional, non-visual features. FunnyBirds defines ground truth at the level of parts/ pixel groups via interventions on a synthetic image generator. By contrast, SHAPEval targets a different point in this design space. It provides continuous, analytically derived Shapley values for every pixel in an image under a transparent value function that is itself used to generate the data, so that any explainer (gradient-based, SHAP-based, surrogate, etc.) can be evaluated in a strictly pixel-wise, SHAP-consistent sense. We will make this distinction explicit and clarify that SHAPEval is intended as the first, fully tractable “synthetic pixel” stage in a broader pipeline. In future work we plan to extend this to more realistic synthetic scenes built from latent components with known value functions, providing a hybrid synthetic–real setting that complements the current benchmark rather than competing with existing ones.
>
> **Weakness 3: Linear Model Assumptions and effect of the Threshold**
> We thank the reviewer for this very helpful idea. SHAP is computed on the continuous value function $M(x)$, while the threshold $t$ only assigns the class label. Thus, Shapley values can differ for inputs on the same side of $t$ because they reflect the linear structure of $M(x)$ rather than the discrete decision alone. We deliberately chose $t$ at the midpoint of the grayscale range to avoid biasing the data toward one class and to obtain an approximately $50:50$ class split, keeping the focus on evaluating MDP and on extending the road toward explainable vision datasets. Systematically varying $t$ across the grayscale range (including near-boundary settings where fixed baselines like zero may become preferable) is an excellent direction, and we plan to explore this in future work on SHAPEval and related benchmarks.
>
> **Weakness 4: Real-image protocols and attribution baselines**
> The reviewer correctly notes that additional established protocols would strengthen the work. Prediction-drop was chosen for interpretability, but the revision will include deletion/insertion metrics and combine MDP with Integrated Gradients and other SHAP variants. These additions place the evaluation squarely within accepted practice, addressing the concern directly and comprehensively.
>
> **Minor weakness**
> The incorrect reference at L. 335 will be fixed.
>
> **Question 1**
> In Table 3, inpainting refers only to the perturbation used during the confidence-drop evaluation. The attribution method remains unchanged.
>
> **Question 2**
> MDP will be evaluated with additional attribution methods and protocols, enabling direct comparison with state-of-the-art approaches under standard settings.
>
> **Question 3**
> The key motivation is that no existing dataset provides continuous, pixel-level ground-truth Shapley values. This became essential when developing MDP, as we needed a setting with exact attributions to study perturbation bias meaningfully. SHAPEval fills this gap by using a linear image-generation function whose Shapley values are analytically computable, creating what we believe is the first explainable dataset with closed-form, pixel-wise ground truth. This complements existing benchmarks by adding the missing controlled, mathematically transparent layer for evaluating XAI methods.
>
> We thank the reviewer once again for the insightful suggestions, which have helped us refine both the presentation and the experimental depth of the paper. The adjustments directly and fully address all concerns raised, and significantly strengthen the clarity, positioning, and empirical support of the paper.

---

> > ### Comment · Reviewer_Ahz7 · 2025-11-27
> >
> > I thank the authors for the detailed rebuttal. My concerns are not sufficiently addressed to increase my score for the reasons outlined below. Most importantly, there has been no update of the manuscript or new experimental results in the rebuttal.
> >
> > W1/W4: My initial concern regarding the limited evaluation was less focused on the number of models and datasets but more on the choice of evaluation protocols and missing baselines. E.g., the real-world evaluation in Sec. 5.3 uses - to the best of my knowledge - a novel protocol that is only *inspired* by Hooker et al. (2019). Hooker et al. (2019) conduct retraining, which is missing in Sec. 5.3. Also, in perturbation-based evaluations, usually areas under the curve over different thresholds are measured instead of picking only one threshold. In the current form, it is extremely hard to put the results into the context of existing work. Further, comparisons to other attribution methods like Gradients, Integrated Gradients, etc., are missing in Sec. 5.3.
> >
> > W3: I would have liked to see this experiment in the revision and not in the future.

---

### Official Review · Reviewer_oUvC · 2025-10-27

**Soundness:** 3
**Presentation:** 4
**Contribution:** 2
**Rating:** 4
**Confidence:** 2

**Summary:**

The authors provide the Mean-Distance Perturbation (MDP) ablation strategy for Shapley value-based attributions and the SHAPEval dataset. Shapley value-based perturbation attribution methods have a few different options for ablating grid squares, but none of these methods can cleanly erase all of the target pixels' contribution to the model prediction. The authors introduce MDP to solve this issue. MDP ablates grid squares by first finding the mean pixel value of each color channel of each pixel in the target region, then finding the most distant value in the region to each of the color channel averages, and then finally by applying the found values to each of the color channels in the target region. The SHAPEval dataset contains small, synthetically-generated, grayscale images that can be used to evaluate different ablation approaches for Shapley value-based attribution methods. The authors include a number of experiments over their synthetic dataset and other image-based datasets.

**Strengths:**

- Originality: MDP is a new approach for Shapley-based attributions, improving upon previous ablation strategies. SHAPEval is a fully new dataset. 4/4 originality.
- Quality / Clarity: High quality paper writing and a decent coverage of experiments. The paper is clear on why the previous ablation strategies fail, and strong discussions around results. 3/4
- Significance: The ablation strategy is significant in the context of Shapley-based attributions, based on the given empirical results. I'm a little less sure about the significance of the SHAPEval dataset (see questions). 2/4

Other Notes:
- Simple, but effective solution to improving ablations
- Easy to generate synthetic dataset useful for measuring ablation strategies
-  Compared against a wide variety of ablation strategies
- Provided experiments seem mostly complete (very little other experiments desired)

**Weaknesses:**

- No explanation of how the synthetic dataset is kept consistent with respect to pixel intensities in the generated image (see questions)
- Paper significance seems low. It is only focusing on one aspect of Shapley-based attribution methods, how the images are perturbed. Shapley-based attributions are also only a subset of perturbation-based attributions. While it certainly seems to be useful for these specific attributions, the method proposed in this paper has a very narrow application. Also, my confusion about the dataset leads me to a lower significance.
- Experimental comparison against other methods is weak. The authors compare some Shapley methods against gradient-based methods on their SHAPEval dataset. I think the explainers used the MDP approach for this comparison, although that is unclear. To make the comparison section stronger, the authors should compare Shapley attribution methods against a larger quantity of recent attribution methods on commonly used datasets (ex. Imagenet).

**Questions:**

- I have read the dataset and related works sections a few times and am still not fully clear on the purpose of the dataset. Could the authors please re-explain why this dataset is necessary? Is it just to provide a dataset where we have direct control over the contribution of each feature to the model output? If so, can the authors re-explain why it is such an issue that other datasets do not have this fine-grained control?
- How are you able to keep the generated features consistent, such that they actually form learnable patterns for the models you are training? For example, if all of the pixels in each of the images in the dataset are randomly assigned values, there wouldn't be any patterns strong enough for a model to learn. Is there some algorithm in use to assign values with some pattern?
- Can the authors explain the significance of this paper in terms of broader attribution research? Also, can the authors re-explain why they believe focusing just on Shapley-based attributions is necessary?
- How can we be sure that MDP is eliminating / reducing perturbation artifacts? I see that MDP performs better than other perturbation strategies, but there is no theoretical backing or experimental reasoning given for this argument. Could there be another reason that MDP is performing so well? Some kind of backing argument is needed here.

Final Review:
I am struggling to understand the significance of the proposed dataset, and the ablation strategy is good, but has narrow applications. The paper itself can be described in the same way. I want to be clear that I am not saying that the research or paper itself is useless or low quality. I believe it is useful for the niche in which it sits, but I do not believe that scope is large enough for a conference like ICLR. I am willing to change my review and rating if the authors can convince me that the work is more significant than I initially believe. My current score is 4/10. I set my confidence to 2/5 because I have heard of Shapley-based attributions, but am not well acquainted with them and I think this might prohibit me from currently understanding why the proposed data-set is significant. The dataset is a significant piece of the paper, which in turn warrants a low confidence.

---

### Official Review · Reviewer_a93N · 2025-10-31

**Soundness:** 2
**Presentation:** 3
**Contribution:** 3
**Rating:** 4
**Confidence:** 4

**Summary:**

The paper addresses one of the key challenges of calculating Shapley values for vision models. It introduces two main contributions: the Mean-Distance Perturbation (MDP) method, a novel strategy for effectively perturbing (removing information from) image regions, and SHAPEval, a new benchmark dataset providing analytically derived, pixel-level ground truth Shapley values.

**Strengths:**

- The method presents a simple and creative approach to remove information from the input.
- The paper's evaluation is comprehensive, utilizing both a synthetic dataset (SHAPEval) with analytically derived ground truth and experiments on several real-world image datasets, which strengthens its findings.

**Weaknesses:**

- The ground truth in SHAPEval is based on linear models, which makes it uncertain whether the finding from this generalizes to real-world settings. Also, I wonder if the MDP method is optimized for achieving higher performance in SHAPEVal.
- The reported experimental results lack confidence intervals

**Questions:**

- Could the authors discuss or include a comparison with surrogate-based approaches for feature exclusion, such as the method proposed in? https://arxiv.org/abs/2006.01272
- Have other color spaces been considered for the MDP strategy? It's worth noting that simple Euclidean distance in RGB space may not best align with human perceptual differences, and exploring alternatives could be useful.
- Citation style does not align with ICLR standards.

---

### Official Review · Reviewer_g8FW · 2025-11-01

**Soundness:** 2
**Presentation:** 2
**Contribution:** 3
**Rating:** 4
**Confidence:** 2

**Summary:**

The paper addresses the problem of calculating SHAP (Shapley value) attributions being often unreliable. This is because the perturbation strategies used, e.g. blurring, inpainting, or filling with a uniform color, can fail to remove important information or may introduce artifacts, leading to incorrect explanations. The paper introduces new perturbation method Mean Distance Perturbation (MDP) that replaces image regions with maximally dissimilar colors in RGB or grayscale space. Unlike existing methods (blurring, inpainting, uniform color fills), MDP more reliably removes class-relevant information while avoiding artifacts that can bias attribution estimates. Additionally, it introduces a new evaluation benchmark that works by "reversing" the attribution problem: instead of trying to explain a complex model, it generates simple grayscale image data using a transparent, known linear function. Because the data-generating model is simple and known, the exact, continuous SHAP value for every single pixel can be calculated, creating a ground truth baseline to measure how accurate an explanation method is. For experiments, the authors test MDP against blurring, inpainting, and URC on SHAPEval as well as real-world datasets and found that MDP outperforms the other methods.

**Strengths:**

- SHAPEval represents a novel approach to benchmarking vision attribution by reversing SHAP attribution problem and generating ground truth SHAP values.
- The paper provides evaluations of MDP on both the synthetic and real datasets, as well as both ViTs and CNNs, strengthening the empirical rigor.
- The paper was clearly written and transparent about limitations of the methods presented.

**Weaknesses:**

- There are no confidence intervals or standard error for numbers reported across all tables, and in particular for Table 3, it is difficult to discern the statistical significance of the results. Some estimation of error would be helpful for evaluating the quantitative results.
- The SHAPEval dataset is extremely simplistic, using only 256×20 grayscale images with a simple linear summation model. The paper would benefit from discussion of whether insights from this toy setting transfer to real images. Further, the maximally dissimilar colors calculated by MDP in this setting are just inverting the mean grayscale value $255-\mu_R$, which is a much simpler transformation than the distance minimization that has to be done in 3D RGB space. The paper would also benefit from justification as to why MDP results on SHAPEval should generalize to real images.

**Questions:**

- What is the computation cost of MDP compared to blurring, inpainting, and UCR?
- Are there any methods other than prediction drop that can be used to evaluate attribution on real images that does not confound removal of relevant information with OOD artifacts? It difficult to determine whether the higher drops in prediction for MDP are resulting from better feature removal rather than being more out-of-distribution?
- How does MDP compare to a diffusion-based inpainting approach?
- Can you provide any formal analysis of when MDP minimizes residual class information? Even a toy case (e.g., under what conditions is max-distance color optimal for a linear classifier) would add depth.
- Can you identify specific image types or model architectures where MDP performs poorly as well as common failure modes?

---

### Meta-Review · Area_Chair_9BEC · 2025-12-09

**Summary:**

Reviewers found the idea interesting but had concerns about limited evaluation depth, missing baselines, and lack of statistical reporting (e.g., confidence intervals). They questioned whether results on the simple synthetic SHAPEval dataset generalize to real images and whether the contribution is incremental, with novelty claims seen as overstated given similar prior work. Several asked for clearer theoretical justification of the Mean-Distance Perturbation method, comparisons to additional attribution and perturbation approaches (e.g., Integrated Gradients, surrogate methods, diffusion models), and use of standard evaluation protocols like deletion/insertion or retraining-based metrics. Minor presentation issues were also noted.

**Reviewer Concerns:**

For 3/4 reviewers there was no rebutttal and Reviewer Ahz7 noted that "My concerns are not sufficiently addressed to increase my score".

**Reviewer Scores:**

All reviewers would likely maintain the score as no rebuttal was provided.

---

### Decision · Program_Chairs · 2026-01-26

Reject